# The Effects of Different Pediatric Drugs and Brushing on the Color Stability of Esthetic Restorative Materials Used in Pediatric Dentistry: An In Vitro Study

**DOI:** 10.3390/children9071026

**Published:** 2022-07-10

**Authors:** Manal Almutairi, Ihab Moussa, Norah Alsaeri, Alhanouf Alqahtani, Shahad Alsulaiman, Maram Alhajri

**Affiliations:** 1Department of Pediatric Dentistry and Orthodontics, College of Dentistry, King Saud University, Riyadh 11545, Saudi Arabia; 2Department of Restorative Dental Sciences, College of Dentistry, King Saud University, Riyadh 11545, Saudi Arabia; ihabmoussa64@hotmail.com; 3Intern, College of Dentistry, King Saud University, Riyadh 11454, Saudi Arabia; norahalsaeri@gmail.com (N.A.); alhanoouuf@gmail.com (A.A.); shoddiei.75@gmail.com (S.A.); maramalhajri4@gmail.com (M.A.)

**Keywords:** restorative materials, pediatric drugs, tooth brushing, composite, spectrophotometer

## Abstract

The aim of the current study is to observe how different pediatric drugs and tooth brushing affect the color stability of different esthetic restorative materials. Three restorative materials (composite, compomer, and glass ionomer cement (GIC)) were each used to produce 72 specimens (10 mm × 2 mm). The specimens were divided into six groups and immersed in distilled water and five different pediatric drugs (amoxicillin, ibuprofen, ventolin, paracetamol, and multivitamins). Each group was divided into two subgroups (brushed and non-brushed). Over the course of two weeks, the specimens were agitated for one minute every eight hours. Color changes in all the specimens were evaluated using a spectrophotometer at 1 and 2 weeks. GIC showed a change in color that was significantly greater than that in all the other materials in each solution, except for those in amoxicillin. After a period of 1 to 2 weeks, the most noticeable change in color was detected in the amoxicillin composite and amoxicillin GIC unbrushed groups, and after 2 weeks, a significant difference was found in the ventolin GIC unbrushed group. The color stability of the restorative materials used in pediatric dentistry can be influenced by using popular liquid pediatric medications. GIC was the least color-stable material when subjected to liquid medications.

## 1. Introduction

The demand for better esthetics with regard to the appearance of dental materials used in pediatric dentistry is ever-increasing [1,2]. Composite resins, polyacid-modified composite resins (compomers), and glass ionomer cement (GIC) are commonly utilized to treat pediatric patients and to produce equivalent esthetic results [3,4,5,6]. These materials are used to help improve deformed teeth by restoring the decayed esthetic zones of posterior and anterior teeth. However, the color stability of the restorative materials is directly proportional to the longevity and acceptance of such restorations [7].

Color stability is the main indicator of whether a restoration has been successful. Staining is a major issue that affects all restorative materials following extended use, and it is caused by both intrinsic and external sources [8,9,10,11,12,13]. Intrinsic color variations may be influenced by various factors, such as the resin matrix composition and the filler particle size and ratio [14]. Inadequate polymerization, the frequent consumption of food and drinks, and medication, including their coloring agents/additives, are all extrinsic causes of the discoloration of teeth [13].

Various liquid drugs are administered to children in order to manage various diseases or symptoms, and some of them are more commonly used than others, such as antibiotics, analgesics, cough medicines, and multivitamins. These drugs are used to improve and protect health as a result of the active ingredients they contain; however, their inactive ingredients may cause undesirable effects [14].

Sugars, acids, buffering agents, and specified coloring substances in the form of oil- and/or water-soluble compounds are all incorporated into these liquid formulations for use in pediatric medicines [15]. The addition of sugars to these medicines masks the unpleasant taste of their active ingredients and thus increases a child’s likelihood of ingesting the medicine [16,17]. Sucrose is a sweetener that is commonly used for such medicated formulations, as it is an ingredient that is easily processed and cost-effective [18]. Fructose and glucose are also added to pediatric liquid medications. These sugars are added to formulations that help to lower the pH level of dental plaque and function as substrates for the fermentation of oral microbiota, which contributes to dental caries. Furthermore, pediatric liquid medication contains certain acids that are added to the solution as buffering agents to maintain chemical stability and to control tonicity and physiological compatibility [19,20].

Despite the fact that previous studies [21,22,23,24,25,26,27] have focused on the acid-degradation effects of pediatric medicines containing sucrose on teeth and restorative materials, there is insufficient evidence that supports the notion that the staining effects of these formulations have been tested on dental materials applicable in pediatric dentistry. Based on the information obtained from consultations with pediatric dentists in Saudi Arabia regarding the drugs that are the most commonly administered to pediatric patients, the following drugs were chosen to be tested in this study: amoxicillin, ibuprofen, paracetamol, ventolin, and multivitamins. Thus, the aim of the present study is to evaluate the effects of different pediatric drugs and tooth brushing on the color stability of three esthetic restorative materials: composite resin, polyacid-modified composite resin (compomer), and glass ionomer cement (GIC).

## 2. Materials and Methods

### 2.1. Specimen Preparation

The present in vitro study was approved by the Institutional Review Board and Ethics Committee of the College of Dentistry Research Center (IRB. No. E-21-6545).

Five liquid drugs commonly administered to children were tested in the present study, and the pH levels were measured using a pH meter (Mettler-Toledo, Columbus, OH, USA) (Table 1).

The details of the materials are shown in Table 2.

Each of the materials provided 72 disk-shaped specimens (10 mm in diameter × 2 mm thick) using a Teflon ring. To avoid air entrapment and voids, a cellulose acetate matrix strip was placed over the ring and held between two 1 mm thick glass slides. Three restorative materials were prepared according to the manufacturer’s instructions [28]. The compomer specimens were polymerized by applying a light-emitting diode (LED) polymerization light (3M ESPE Dental, Dublin, Ireland) of 1200 mW/cm^2^ to each surface for 20 s, with the tip of the light unit on the glass slide (0 mm away from the specimen). During the preparation of the specimens for composite resin, the material used for each specimen was placed in a Teflon ring and covered with a mylar strip, which was followed by light curing for 20 s using an LED light-curing unit. When preparing the specimens for the GIC, each specimen was activated and mixed in an amalgamator (Gnatus Amalga mix 2) for 10 s, according to the manufacturer’s instructions. The specimens were produced by inserting the necessary materials into a mold with a carrier and then by curing the materials against the mylar strip to achieve the smoothest surfaces. The material was light-cured for 20 s. Following the polymerization process, the specimens were polished with aluminum oxide disks (Sof-Lex, 3M ESPE, St. Paul, MN, USA) and an electric handpiece operated at 15,000 rpm for 10 s on each disk (coarse, medium, fine, and superfine). To complete the polymerization process, all the specimens were stored in distilled water at 37 °C for 24 h [29].

### 2.2. Subgrouping of the Specimens

A total of 216 specimens were produced using three different types of restorative materials (72 from each). The specimens of each restorative material were randomly divided into six solution groups (n = 12) based on the pediatric drug formulations to be tested. The groups were as follows: Group 1—amoxicillin; Group 2—ventolin; Group 3—multivitamins; Group 4—ibuprofen; Group 5—paracetamol; and Group 6—distilled water. The control solution was distilled water (pH 5.7). Each group (n = 6) was divided into two subgroups (brushed and unbrushed). The G*Power software tool version 3.1.9.2 was used to compute the sample size, which was set to a minimum of 6 specimens per group: power 0.95, = 0.05, effect side = 0.3 (medium to large).

### 2.3. Color-Change Measurement and Brushing Cycles

After the specimens were polished, they were rinsed with distilled water for 5 s and dried with tissue paper before being measured with LabScan XE spectrophotometer (HunterLab, Reston, VA, USA) to determine the baseline color values. The spectrophotometer was calibrated with its own calibration device prior to obtaining the color measurements of the specimens, which were taken at the center of each specimen. A clinical spectrophotometer was used to obtain whole color measurements with a CIEDE2000 color system against a standard white background using D65 standard light. Measurements were obtained three times for each specimen, according to the CIE *L***a***b** system, and the average was recorded. The CIELAB system is a chromatic value color space that measures chroma and values via three coordinates: *L** represents brightness or lightness (value); *a** and *b** serve as numeric correlates for both hue and chroma. The *a** and *b** values represent the positions on red/green and yellow/blue axes, respectively [30].

Subsequently, the specimens were immersed (n = 12; for each material) in 5 different 10 mL undiluted pediatric liquids in test tubes and agitated for 1 min every 8 h for 2 weeks. The solutions were replaced on a daily basis. Once a week, the antibiotic was prepared and stored in a refrigerator. Between immersion intervals, the specimens were stored in distilled water. To standardize temperature, a thermometer (JiangSu YuYue Medical, Danyang, China) was used to test the temperatures of all the solutions (room temperature). The brushing subgroups were brushed once a day with an Oral-B Genius Pro 9000 electric toothbrush (Braun, Melsungen, Germany) using a fluoride-free toothpaste (R.O.C.S Kids Fruity Cone, Tallinn, Estonia). A total of 2 mL of toothpaste was applied to the surfaces of the evaluated materials to simulate home-application procedures. In the “continuous”mode, the same operator brushed each specimen 40 times with a standardized force of 2 N. This number was calculated with the expectation that each tooth would be brushed for 10 s over a duration of 2 min [31]. Following the brushing process, the specimen was rinsed under running water and stored in distilled water until the next application.

The color of the specimens was calculated using the spectrophotometer once the immersion time was completed, as previously described. The mean values Δ*E* were calculated and recorded after obtaining three measurements for each specimen. The changes in color were computed between the baseline measurements, and the measurements were obtained at 7 (*E*1) and 14 days (*E*2). The measurements were obtained using the CIEDE2000 (Δ*E*) system. Δ*E* was calculated using the following formula [28]:Δ*E* (*L***a***b**) = ([Δ*L**]2 + [Δ*a**]2 + [Δ*b**]2)^1/2^
where Δ*L** is the difference between the *L** values, Δ*a** is the difference between the *a** values, and Δ*b** is the difference between the *b** values.

### 2.4. Statistical Analysis

Statistical analyses concerning chromatic value space were performed on the data obtained, which measure the value of the chroma. SPSS software version 26.0 was used to perform statistical analyses (IBM Inc., Chicago, IL, USA). To compare the mean values of Δ*E*1 and Δ*E*2 considering the three factors (type of material, type of solution, and brushing status), a three-way analysis of variance was used, followed by a one-way analysis of variance and Tukey’s post hoc test. The t-test was used to compare the mean values of the brushing and non-brushing groups. The results were significant when *p* < 0.05.

## 3. Results

A comparison of the mean values of Δ*E*1 and Δ*E*2 among the six types of solutions in each of the three restorative materials is described in Table 3.

For the compomer material, the mean values of Δ*E*1 were not statistically significantly different for the different solutions, whereas for Δ*E*2, there was a statistically significant difference in the mean values among the six types of solutions (amoxicillin, ibuprofen, ventolin, paracetamol, multivitamin, and water) (*p* < 0.001). The post hoc multiple comparison test showed that the mean Δ*E*2 value was higher for the ventolin solution (6.48) and was found to be significantly higher than the mean values of the other five solutions (2.06, 2.02, 2.21, 2.30, and 1.75, respectively). In the case of the composite material, the mean values of Δ*E*1 and Δ*E*2 were not statistically significantly different among all six types of solutions (1.14 and 1.28, 1.28 and 1.22, 1.07 and 1.04, 1.05 and 1.13, 1.03 and 1.33, and 0.72 and 0.78, respectively).

The GIC material showed a statistically significant difference for the mean values of both Δ*E*1 and Δ*E*2 (*p* = 0.001 and *p* < 0.0001). The post hoc multiple comparison test showed that the mean Δ*E*1 value was significantly higher for ventolin, followed by paracetamol (6.27, 5.29). Additionally, for Δ*E*2, the mean values of the Ventolin solution (6.26) were significantly higher than for the other five solutions (3.03, 4.34, 3.72, 5.92, and 6.18, respectively).

Comparing the mean values of Δ*E*1 and Δ*E*2 among the three restorative materials in each of the six types of solutions, the value for the GIC material is significantly higher when compared with those of the other two types of materials (compomer and composite) in each of the six types of solutions, except in the amoxicillin solution, in which there is no statistically significant difference in the mean values of Δ*E*2 among the three types of restorative materials (Table 3).

In regard to the effect of tooth brushing (Table 4), a comparison of the mean values of Δ*E*1 and Δ*E*2 between the samples that were brushed and unbrushed with the use of each of the six solutions in each of the three restorative materials shows statistically significant differences in the mean values of Δ*E*1 for the amoxicillin solution in the composite (t = 3.141, *p* = 0.010) and GIC (t = 4.82, *p* < 0.0001) materials, where the mean value of Δ*E*1 was significantly higher in the samples that were unbrushed. Additionally, there are statistically significant differences in the mean values of Δ*E*2 for the ventolin solution in the compomer material (t = 4.156, *p* = 0.002) and for the paracetamol solution in the GIC material (t = 2.355; *p* = 0.040) in the brushed samples, and for the amoxicillin solution in the composite material (t = 3.041, *p* = 0.012), for the amoxicillin solution in the GIC material (t = 6.342, *p* = 0.0001), and for the ventolin solution in the GIC material (t = 3.351, *p* = 0.007) in the unbrushed samples (Table 4).

The three-way analysis of variance values obtained for Δ*E*1 and Δ*E*2 by using three types of materials, six types of solutions, and two categories of brushing showed highly statistically significant differences in the overall model (*p* < 0.0001), for the material (*p* < 0.0001) and for the solution (*p* < 0.0001), and was not statistically significant for brushing. Out of the three two-way terms (material * solution, material * brushing, and solution * brushing), only two terms were statistically significant. Material * brushing was not statistically significant; however, the three-way term (material * solution * brushing) in Δ*E*1 and Δ*E*2 was statistically significant. This indicates that there is an interaction between the type of material and solution and between the solution and brushing but no interaction between material and brushing (Table 5).

## 4. Discussion

A thorough review of the literature revealed that only a limited number of studies have examined the effects of pediatric liquid medications and brushing on the color stability of pediatric restorative materials [14,20,32,33]. In contrast to the previous studies, the present study’s immersion period was extended to 2 weeks at intervals of 8 h in order to account for the clinical scenario in which the children are regularly affected by the most common pediatric drugs. Therefore, the purpose of this study was to assess the impact of teeth brushing on the color stability of three aesthetic restorative materials, namely composite resin, compomer, and GIC, following exposure to the most commonly prescribed pediatric medications for one to two weeks.

In pediatric dentistry, the long-term and/or chronic use of prescription pediatric medicines may cause a lower plaque pH, resulting in cariogenic and erosive potentials [34]. In addition, the high viscosity of liquid medicines and coloring agents used in them affect the color stability of these restorative materials [33]. However, the pH information for the pediatric liquid drugs selected for this study was not provided by the manufacturers. As a result, a digital pH meter was used to determine the pH levels of each solution prior to the investigation. Amoxicillin, paracetamol, ibuprofen, Ventolin, and the multivitamins had pH values of 6.19, 5.09, 4.60, 3.72, and 3.88, respectively. All of these values are below the critical pH of 5.5, except for that of amoxicillin. Tooth brushing is an important practice in basic dental hygiene. Tooth brushing causes considerable dental wear. The factors of brushing technique, time, and force are all important for maintaining teeth and in dental restorations [35].

As a result of the reasons mentioned above, the first part of the study aimed to evaluate the color stability of three esthetic restorative materials—composite resin, compomer, and GIC—when subjected to different pediatric liquid drugs. In the present study, we used a spectrophotometer to minimize bias due to the color sensitivity of humans, specifically the investigators, thus providing various additional benefits, such as repeatability, sensitivity, and objectivity. To accurately simulate reality, the specimens were immersed daily in a pediatric liquid for 1 min every 8 h, and for the rest of the day, they were stored in distilled water in a manner similar to that presented in the study conducted by Yildirim and Uslu [14]. Instead of artificial saliva, distilled water was utilized to store the specimens. Turssi and co-workers demonstrated that resin-based materials stored in either distilled water or artificial saliva had a similar micromorphology [36]. To standardize the abrasive impact of tooth brushing, an electric toothbrush and fluoride-free toothpaste were used.

In the results obtained in the present study, we found that GIC showed a significantly greater change in color when compared to the other two types of materials in each of the six types of solutions, except in amoxicillin. This part of our result is consistent with the results obtained from a study conducted by Adusumilli et al., who studied the color stability of esthetic restorative materials used in pediatric dentistry (GIC and Giomer) and who found that GIC had the greatest color change when compared to the Giomer in all the immersion media and among all the immersion regimes [37]. Kalampalikis et al. and Chhabra C et al. stated that the lack of color stability in conventional GIC could be caused by the polyacid content of the material, which relates to the degradation of metal polyacrylate salts [38,39]. Additionally, there are many reasons for the susceptibility of GIC to staining, including its porosity, dehydration after setting and drying, and microcracks, which allow for discoloration and staining to occur. Shalan et al. determined that GIC subgroups showed a higher susceptibility to discoloration than compomer and explained that fluoride-releasing materials released a considerable number of ions when subjected to pH variations, thus resulting in a high ionic exchange rate, which ultimately resulted in a change in color [34]. This explanation is supported by previous studies conducted by Forss H and Williams JA [40,41], who stated that the fluoride-releasing materials release more ions in the presence of pH variations that could lead to less color stability when compared to composite resins. Additionally, Hotwani et al. stated that hydrophobic substances, such as resin composite, are assumed to possess greater color stability and stain resistance compared to hydrophilic substances such as GIC and compomer [30].

In contrast to our findings, previous studies evaluating the color stability of pediatric restorative materials showed that the composite is the least color-stable material [12,20,28,33]. Our results show that a significant change in color in the amoxicillin composite, amoxicillin GIC, and ventolin GIC groups, which is inconsistent with the results of previous studies, might be due to a variety of factors, including the presence or absence of the brushing component, the difference in the experiment’s duration, the medications utilized and their pH levels, the brands of the materials used, or the surface roughness.

The second part of our study aimed to evaluate the influence of the different drugs on the color stability of restorative materials. We determined that the amoxicillin composite, amoxicillin GIC, and Ventolin GIC groups all showed significant changes in color. Our findings confirm those presented by Kale et al., who reported that the staining ability of the amoxicillin + clavulanic acid group was second only to the metronidazole group. The high viscosity of these liquid medications, according to Kale and colleagues, permits them to stay on the tooth’s surface for a longer period of time. Furthermore, a low salivary clearance rate results in more unfavorable consequences. The approved coloring ingredients in these liquid medications are also absorbed and adsorbed. The researchers observed that ibuprofen is associated with the weakest staining ability [20], which is a result similar to that presented by Faghihi et al., who showed that the least observable ΔE in the resin-reinforced GIC group was associated with ibuprofen [32]. These results support those presented in another study that determined that the ACTIVA-KIDS ibuprofen group had the weakest color-changing ability, while iron supplements had the strongest staining ability of all the drugs that were assessed [33].

The third part of our study evaluated the effect of tooth brushing on the color stability of three esthetic restorative materials. In the present study, following 1 and 2 weeks of exposure to the most commonly used pediatric medicines, the most noticeable color change was detected in the amoxicillin composite and amoxicillin GIC unbrushed groups, and after 2 weeks, a significant difference was observed in the ventolin GIC unbrushed group. In a similar study conducted by Yldrm et al., the non-brushing group using Floradix compomer had the highest *E* value at week 2. Yldrm and colleagues observed that tooth brushing significantly increased the color stability of esthetic restorative materials [14]. Color change is influenced by the ingredients in pediatric medications, and the discoloration effect of drug solutions on restorative materials is influenced by the material’s composition, the type of pigments contained in the solutions, and exposure time [14]. Shalan et al. investigated the effects of colored beverages on several esthetic restorative materials in primary teeth and discovered that the tooth-brushing subgroups using the same materials and drinking the same beverage exhibited less color change than the non-tooth-brushing subgroups [34]. This result corresponds with that presented by Bezgin et al., who investigated the effect of tooth brushing on color changes in esthetic restorative materials and who found that brushing significantly lessened the change in color of teeth [12].

However, this study has certain limitations, such as the short period of the study (2 weeks), the small number of materials and the roles of saliva, the oral environment, or oral clearance of liquid medication formulations, which were reflected in the in vitro experimental conditions and may not be adequate to realistically represent the conditions of the oral environment. The salivary content and buffering capacity, the structural features, the compositions of the medications, and utilizing the drug at irregular intervals may influence the change in the color of teeth when using pediatric restorative materials. Furthermore, since the sample size utilized in the current study was small, further research using larger sample sizes is required.

## 5. Conclusions

Within the limitations of this study, the following conclusions were drawn: the color stability of the restorative materials used in pediatric dentistry can be influenced by using popular liquid pediatric medications, and GIC proved to be the least color-stable material when subjected to liquid pediatric medications. These results can inform child healthcare providers, pediatric dentists, and parents of the risk of tooth surface/restoration discoloration. Further studies using in vivo study designs are required to support the results of this study.

## Figures and Tables

**Table 1 children-09-01026-t001:** Pediatric liquid drugs used in this study.

Generic Name	Brand Name	Therapeutic Class	pH
Amoxicillin	Neomox	Antibiotics	6.19
Paracetamol	Fevadol	Analgesics	5.09
Ibuprofen	Nurofen	Analgesics	4.60
Ventolin	Ventolin	Bronchodilator	3.72
Multivitamins	Sanovit	Multivitamins	3.88

**Table 2 children-09-01026-t002:** Restorative materials used in the study.

Product	Material Type	Mixing	Curing	Manufacturer
Dyract XP	Polyacid-modified composite resin (compomer)	N\A	Light cure for 20 s	Dentsply DeTrey, GmbH, Germany
Tetric N-Ceram	Composite resin	N\A	Light cure for 20 s	Ivoclar Vivadent AG, Liechtenstein
Fuji II LC Capsules	Resin-modified glass ionomer	10 s	Light cure for 20 s	GC Corporation, Tokyo, Japan

**Table 3 children-09-01026-t003:** Mean and standard deviation of Δ*E* values of the restorative materials tested with pediatric drugs.

Time	Solutions	Restorative Material					
		Compomer (n = 72)		Composite (n = 72)		GIC (n = 72)	
		Mean ± SD	*p*-Value	Mean ± SD	*p*-Value	Mean ± SD	*p*-Value
Week 1	Amoxicillin	1.23 ± 0.81	0.250	1.14 ± 0.51	0.540	3.16 ± 2.72	0.001
	Ibuprofen	2.39 ± 1.34		1.28 ± 1.31		4.76 ± 1.65	
	Ventolin	1.90 ± 1.81		1.07 ± 0.44		6.27 ± 1.05	
	Paracetamol	1.55 ± 0.48		1.05 ± 0.59		5.29 ± 1.57	
	Multivitamin	1.64 ± 0.55		1.03 ± 0.48		4.35 ± 0.99	
	Water	1.71 ± 1.33		0.72 ± 0.42		4.87 ± 1.70	
Week 2	Amoxicillin	2.06 ± 2.29	<0.0001	1.28 ± 0.82	0.537	3.03 ± 2.21	<0.0001
	Ibuprofen	2.02 ± 1.55		1.22 ± 1.39		4.34 ± 1.68	
	Ventolin	6.48 ± 2.76		1.04 ± 0.44		6.26 ± 2.06	
	Paracetamol	2.21 ± 0.78		1.13 ± 0.56		3.72 ± 1.19	
	Multivitamin	2.30 ± 0.83		1.33 ± 0.44		5.92 ± 1.43	
	Water	1.75 ± 1.38		0.78 ± 0.44		6.18 ± 2.07	

SD = standard deviation; GIC = glass ionomer cement; statistically significant at *p* < 0.05.

**Table 4 children-09-01026-t004:** Effect of pediatric drugs and brushing on color change in the restorations.

Time	Solutions	Restorative Material								
		Compomer (n = 72)			Composite (n = 72)			GIC (n = 72)		
		Brushed Mean ± SD	Unbrushed Mean ± SD	*p*-Value	Brushed Mean ± SD	Unbrushed Mean ± SD	*p*-Value	Brushed Mean ± SD	Unbrushed Mean ± SD	*p*-Value
Week 1	Amoxicillin	0.94 ± 0.53	1.51 ± 0.99	0.242	0.79 ± 0.40	1.48 ± 0.36	0.010	0.69 ± 0.23	5.51 ± 1.51	0.0001
	Ibuprofen	2.08 ± 1.40	2.71 ± 1.33	0.443	1.57 ± 1.86	0.98 ± 0.35	0.463	5.58 ± 1.99	3.94 ± 0.61	0.082
	Ventolin	2.58 ± 2.41	1.21 ± 0.49	0.202	1.23 ± 0.60	0.91 ± 0.10	0.226	5.73 ± 0.52	6.80 ± 1.21	0.075
	Paracetamol	1.50 ± 0.61	1.61 ± 0.36	0.712	0.87 ± 0.08	1.23 ± 0.83	0.315	5.98 ± 0.95	4.61 ± 1.83	0.135
	Multivitamin	1.68 ± 0.63	1.59 ± 0.53	0.794	1.02 ± 0.51	1.03 ± 0.50	0.973	4.46 ± 0.91	4.24 ± 1.14	0.719
	Water	1.97 ± 1.50	1.45 ± 1.23	0.526	0.75 ± 0.39	0.69 ± 0.49	0.819	4.63 ± 2.32	5.10 ± 0.89	0.653
Week 2	Amoxicillin	1.20 ± 0.72	2.91 ± 3.04	0.209	0.73 ± 0.59	1.82 ± 0.65	0.012	1.13 ± 0.20	4.92 ± 1.45	0.0001
	Ibuprofen	1.75 ± 1.71	2.28 ± 1.48	0.578	1.61 ± 1.85	0.83 ± 0.70	0.357	4.72 ± 2.31	3.96 ± 0.70	0.458
	Ventolin	6.10 ± 2.84	1.21 ± 0.49	0.002	0.99 ± 1.08	1.08 ± 0.49	0.856	4.82 ± 0.91	7.69 ± 1.89	0.007
	Paracetamol	2.38 ± 0.88	2.04 ± 0.70	0.476	0.85 ± 0.55	1.40 ± 0.45	0.087	4.40 ± 1.03	3.04 ± 0.97	0.040
	Multivitamin	2.02 ± 0.59	2.58 ± 0.99	0.261	1.32 ± 0.56	1.33 ± 0.33	0.971	5.78 ± 1.86	6.07 ± 1.00	0.743
	Water	2.07 ± 1.57	1.42 ± 1.20	0.439	0.95 ± 0.47	0.60 ± 0.36	0.178	5.99 ± 2.78	6.30 ± 1.26	0.809

SD = standard deviation; GIC = glass ionomer cement; statistically significant at *p* < 0.05.

**Table 5 children-09-01026-t005:** Interaction among Δ*E* values in the three-way analysis of variance.

Time	Variable	Type III Sum of Squares	df	Mean Square	F	*p*-Value
Week 1	Corrected Model	742.540	35	21.215	17.832	0.000
	Intercept	1370.880	1	1370.880	1152.278	0.000
	Material	566.109	2	283.054	237.918	0.000
	Solution	33.662	5	6.732	5.659	0.000
	Brushing	1.072	1	1.072	0.901	0.344
	Material * Solution	43.137	10	4.314	3.626	0.000
	Material * Brushing	4.027	2	2.013	1.692	0.187
	Solution * Brushing	39.854	5	7.971	6.700	0.000
	Material * Solution * Brushing	54.678	10	5.468	4.596	0.000
	Error	214.148	180	1.190		
	Total	2327.568	216			
Week 2	Corrected Model	929.239	35	26.550	13.205	0.000
	Intercept	1873.786	1	1873.786	931.970	0.000
	Material	516.691	2	258.346	128.494	0.000
	Solution	143.304	5	28.661	14.255	0.000
	Brushing	11.727	1	11.727	5.833	0.017
	Material * Solution	173.047	10	17.305	8.607	0.000
	Material * Brushing	5.323	2	2.662	1.324	0.269
	Solution * Brushing	48.848	5	9.770	4.859	0.000
	Material * Solution * Brushing	30.298	10	3.030	1.507	0.140
	Error	361.902	180	2.011		
	Total	3164.926	216			

df = degrees of freedom; statistically significant at *p* < 0.05.

## Data Availability

The datasets used and analyzed during the current study are available from the corresponding author upon reasonable request.

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
