# Peer review of "The Effects of Different Pediatric Drugs and Brushing on the Color Stability of Esthetic Restorative Materials Used in Pediatric Dentistry: An In Vitro Study"

_children, 2022, doi:10.3390/children9071026_

Round 1
Reviewer 1 Report
Dear authors,
I had the opportunity of revising this interesting manuscript.
I list here some observations:
ABSTRACT
Line 15: remove the “- “
Line 26: I think you can use the words “Influenced by” instead of “affected by”, so as to make the concept better
INTRODUCTION
in my opinion the introduction should be expanded, for example by explaining which drugs are the most used by children, and therefore highlighting the need to carry out this study on these drugs, maybe with a literature mention.
M&M
why was glycerin not used as a polymerization aid? as reported for example by “Bergmann P, Noack MJ, Roulet JF (1991) Marginal adaptation with glass-ceramic inlays adhesively luted with glycerine gel. Quintessence Int 22:739–744 “
RESULTS
the results are well described and associated with relevant bibliographic references
the limitations of the study were well explained
CONCLUSIONS
the conclusions are appropriate. I think it is better to write conclusions in a discursive way, in order to make them understand the importance of correct information for professionals about these issues and the need to carry out further studies to verify these records in vivo
Best Regards
Author Response
The authors would like to thank the area editor and the reviewers for their precious time and invaluable comments. We have carefully addressed all the comments. The corresponding changes and refinements made in the revised paper are summarized in our response below.
Reviewers comment: English language and style are fine/minor spell check required
Authors’ Response: Thank you! We found your comments extremely helpful and have revised accordingly
Reviewer #1: Line 15: remove the “- “
Authors’ Response: Thank you so much for your comment. We totally agree with your suggestion.
Reviewer #1: Line 26: I think you can use the words “Influenced by” instead of “affected by”, so as to make the concept better
Authors’ Response: Thank you so much for your comment. We fully agree with your comment. The new sentence is now written “… Influenced by ….”
Reviewer #1: In my opinion the introduction should be expanded, for example by explaining which drugs are the most used by children, and therefore highlighting the need to carry out this study on these drugs, maybe with a literature mention.
Authors’ Response: Thank you so much for your comments. Introduction section is thoroughly rewrite considering to your valuable suggestion.
Reviewer #1: why was glycerin not used as a polymerization aid? as reported for example by “Bergmann P, Noack MJ, Roulet JF (1991) Marginal adaptation with glass-ceramic inlays adhesively luted with glycerine gel. Quintessence Int 22:739–744 “
Authors’ Response: We thank the reviewer for this question. There are two reasons why we did not use glycerin in our study.
First and main, the use of glycerine prior to light curing is to inhibit the binding between free radicals and oxygen on the resin composite's surface. Because glycerine is a more stable solution than oxygen, it can be utilized as a protective gel in the polymerization of resin-based composites."’’ Becker LC. Safety Assessment of Glycerin as Used in Cosmetics. Int J Toxicol. 2014’’. In our study, only one layer of restoration with a thickness of 2 mm and used a cellulose acetate matrix strip. However, Marigo's research (2019) found that providing glycerine before light curing had no significant effect on the surface hardness of the tooth filling.’’ Marigo L, Nocca G, Fiorenzano G, Callà C, Castagnola R, Cordaro M, et al. Influences of Different Air-Inhibition Coatings on Monomer Release, Microhardness, and Color Stability of Two Composite Materials. Biomed Res Int. 2019;2019.’’
The second justification is that, similar to previous studies, we would like to asset color change without any effect of physical barriers on the surface of restoration.’’ Yildirim S, Uslu YS. Effects of different pediatric drugs and toothbrushing on color change of restorative materials used in pediatric dentistry. Niger J Clin Pract. 2020;23(5):610-618; Faghihi T, Heidarzadeh Z, Jafari K, Farhoudi I, Hekmatfar S. An experimental study on the effect of four pediatric drug types on color stability in different tooth-colored restorative materials. Dent Res J (Isfahan). 2021; 18:75; Gurdogan Guler EB, Bayrak GD, Unsal M, Selvi Kuvvetli S. Effect of pediatric multivitamin syrups and effervescent tablets on the surface microhardness and roughness of restorative materials: An in vitro study. J Dent Sci. 2021;16(1):311-317’’
Reviewer #1: the results are well described and associated with relevant bibliographic references the limitations of the study were well explained
Authors’ Response: Thank you so much for your comments.
Reviewer #1: the conclusions are appropriate. I think it is better to write conclusions in a discursive way, in order to make them understand the importance of correct information for professionals about these issues and the need to carry out further studies to verify these records in vivo
Authors’ Response: Thank you so much for your suggestion. Conclusion section is also revised for better interpretation

Reviewer 2 Report
Please see the attachment
Minor Observations
Language review was required
In line 37 delete the last word (used).
In lines 61-62: The present in vitro study was approved by the Institutional Review Board and Eth- ics Committee of …… Please do not separate this word.
In lines 117-118 please writing in upper and lower case letters
Background Observations
Introduction
Lines 45-46, the paragraph is very unspecific. It would be important to specify what type of formulations. Need redaction.
In line 58 the authors need specified …… Glass Ionomers Cement (GIC), ……. the acronym should be placed in this paragraph, because it´s the most important, it´s the aim of the study.
Materials and Methods
I line 64 the authors write …….. A2 color shades……., I don’t know if the authors used the VITA Colour Guide or what guide they used, they need specify.
In table 1 the authors put the pH of each substance and did not specify where they got the data from.
In all the tables the authors need add the footnotes.
In line 74 please you need specify the reference of the techniques used and of the supplier must be included.
In line 143 The authors write ….. The confidence level was set as 95% …. It´s not usual to put this phrase, generally was used: the significance value will be p<0.05.
Results
In line 146 I suggested the follow redaction: …. each of the three restorative materials was described in table 3.
In line 147 describe which of the materials described in table 2 is the compomer material or analyse the wording.
In lines 150-152 please It would be very important to add images
In table 3 and 4 please place the footnotes such as SD= standard deviation; GIC = Glass Ionomer Cement; etc.
In table 4 we need see de P values for each comparison’s.
In this section it would be interesting to add a table 5 with the statistical model to be tested.
Discussion
The discussion begins with the findings of the research group and them the discussion with other reports. As well, as the authors' analysis of why they believe these findings occurred.
Although some of the results are discussed throughout the text, the outcome of the research is not clear.
In the paragraph between lines 252-254 I found one of the most important findings.
In lines 257-258 are the principal conclusion
Conclusions
Lines 293-294 You can’t declare that, because in yours results the group describe the follow sentence: …….. but no interaction between material and brushing……
References
Only 10 of 40 references are less than 5 years old (25%).
There are in PUBMED 82 references in the last 5 years.

Author Response
The authors would like to thank the area editor and the reviewers for their precious time and invaluable comments. We have carefully addressed all the comments. The corresponding changes and refinements made in the revised paper are summarized in our response below.
Reviewer #2: Language review was required
Authors’ Response: Thank you for pointing this out and we all spelling and grammatical errors now checked by English language editing.
Reviewer #2: In line 37 delete the last word (used).
Authors’ Response: Thank you so much for your comment. We totally agree with your suggestion.
Reviewer #2: Lines 45-46, the paragraph is very unspecific. It would be important to specify what type of formulations. Need redaction.
Authors’ Response: Thank you so much for your comment. We fully agree with your comment. The new paragraph is now written in introduction “Sugars, acids, buffering agents, and specified coloring substances in the form of oil and/or water-soluble compounds are all incorporated in these liquid formulations [16]. Adding sugars to it this would mask the taste of its unpleasant active ingredient and hence, help to gain the child’s compliance. [17-18] Sucrose is the commonly adjoined sweetener for such medicated formulations as it is an easily processed substance, cost-effective [19]. Fructose and glucose are also added to the pediatric liquid mediacations. These added sugars in the formulations helps to lower the pH of dental plaque and functions as a substrate for oral microbiota fermentation, which contributes to dental caries. Pediatric liquid medicaments also contain certain acids, which are added as buffering agents to the solution to maintain chemical stability, control tonicity, and the physiological compatibility. ]20,21[.”
Reviewer #2: In line 58 the authors need specified …… Glass Ionomers Cement (GIC), ……. the acronym should be placed in this paragraph, because it´s the most important, it´s the aim of the study.
Authors’ Response: Thank you so much for your comment. We fully agree with your comment. The new sentence is now written as “…..Glass Ionomers Cement (GIC).”
Reviewer #2: In lines 61-62: The present in vitro study was approved by the Institutional Review Board and Eth- ics Committee of …… Please do not separate this word.
Authors’ Response: Thank you so much for your comment. We totally agree with your suggestion.
Reviewer #2: In line 64 the authors write ……. A2 color shades……., I don’t know if the authors used the VITA Colour Guide or what guide they used, they need specify.
Authors’ Response: Thank you so much for your comment. We add now name of guide used in selection of shade. The new sentences is now written ‘’Three commercially available tooth-colored restorative materials were tested using A2 color shades from the manufacturer's shade guide: Resin-modified glass ionomer (RMGIC) (Fuji II LC Capsules) and Compomer (Dyract XP) used the VITA Colour Guide, whereas composite resin (Tetric N-Ceram) used the Tetric N-family shade guide.’’
Reviewer #2: In table 1 the authors put the pH of each substance and did not specify where they got the data from.
Authors’ Response: Thank you so much for your comment and we fully agree with your comment. We now clarified this in Material and methods by add new sentence “and pH was measured using the pH meter (Mettler-Toledo, LLC 1900 Polaris Parkway Columbus, USA) [Table 1].”
Reviewer #2: In line 74 please you need specify the reference of the techniques used and of the supplier must be included.
Authors’ Response: Thank you so much for taking the time to give a comment. We completely agree with your suggestion, and we've added a reference for the procedures, however the supplier information is already included in the text and tables.
Reviewer #2: In lines 117-118 please writing in upper and lower case letters
Authors’ Response: Thank you so much for your suggestion. We change now to upper and lower case letters
Reviewer #2: In all the tables the authors need add the footnotes.
Authors’ Response: Thank you so much for your comment. We fully agree with your comment. Only tables 3 and 4 can have footnotes applied.
Reviewer #2: In table 3 and 4 please place the footnotes such as SD= standard deviation; GIC = Glass Ionomer Cement; etc.
Authors’ Response: Thank you so much for your comment. We have taken reviewer’s comment in full consideration and add the footnotes to all the tables. We add SD= standard deviation; GIC = Glass Ionomer Cement and title p in both tables 3 &4.
Table 3. Mean and standard deviation of Δ? values of tested restorative materials with pediatric drugs
|
Time |
Solutions |
Restorative material |
|
|||||
|
Compomer |
Composite |
GIC |
||||||
|
Mean ± SD |
P-Value |
Mean ± SD |
P-Value |
Mean ± SD |
P-Value |
|||
|
Week 1 |
Amoxicillin Ibuprofen Ventolin Paracetamol Multivitamin Water |
1.23 ± 0.81 2.39 ± 1.34 1.90 ± 1.81 1.55 ± 0.48 1.64 ± 0.55 1.71 ± 1.33 |
0.250 |
1.14± 0.51 1.28 ± 1.31 1.07 ± 0.44 1.05 ± 0.59 1.03 ± 0.48 0.72 ± 0.42 |
0.540 |
3.16 ± 2.72 4.76 ± 1.65 6.27 ± 1.05 5.29 ± 1.57 4.35 ± 0.99 4.87 ±1.70 |
0.001 |
|
|
Week 2 |
Amoxicillin Ibuprofen Ventolin Paracetamol Multivitamin Water |
2.06 ± 2.29 2.02 ± 1.55 6.48 ± 2.76 2.21 ± 0.78 2.30 ± 0.83 1.75 ± 1.38 |
<0.0001 |
1.28 ± 0.82 1.22 ± 1.39 1.04 ±0.44 1.13± 0.56 1.33 ± 0.44 0.78 ± 0.44 |
0.537 |
3.03 ± 2.21 4.34 ± 1.68 6.26 ± 2.06 3.72 ± 1.19 5.92 ± 1.43 6.18 ± 2.07 |
<0.0001 |
|
SD= Standard deviation; GIC = Glass Ionomer Cement; Statistically significant at P≤ 0.05.
